# Implementation of Urban Green Infrastructures in Peri-Urban Areas: A Case Study of Climate Change Mitigation in Madrid

**María Teresa Gómez-Villarino** [1,*] **, Miguel Gómez Villarino** [2] **and Luis Ruiz-Garcia** [1]

1   School of Agricultural, Food and Biosystems Engineering, Universidad Politécnica de Madrid, 28040 Madrid, Spain; luis.ruiz@upm.es
2   Independent Consultant, 28036 Madrid, Spain; miguelgvillarino@gmail.com
*   Correspondence: teresa.gomez.villarino@upm.es

**Abstract:** Urban areas are critical points that contribute to global warming and are also affected by climate change. One of the measures to move toward urban sustainability and to reduce the effects of climate change is the development of urban green infrastructures. Urban green infrastructures (UGIs) are being increasingly recognized as key providers of ecosystem services in cities, but there is still a lack of support from urban planners. We highlight the potential of urban green infrastructures for sustainable urban planning based on its capacity to mitigate climate change This paper studies the $CO_2$ mitigation potential through a multi-intervention (agricultural and forestry) local case in the peri-urban surroundings of a big European city such as Madrid. We consider two inseparable aspects: the amount of atmospheric $CO_{2\text{-eq}}$ reduced through direct carbon uptake of the UGI and also the emission of greenhouse gases due to its implementation and maintenance. The analysis carried out has shown the benefits of urban green infrastructures and their contribution to the mitigation of climate change. The results demonstrate that the absorption capacity of the new urban green infrastructure is much greater than its ecological footprint. Therefore, it contributes to the mitigation of emissions from other urban activities, thus improving urban sustainability.

**Keywords:** climate change; ecological footprint; recovery of degraded areas; peri-urban; urban green infrastructures; urban agriculture; urban forestry

## 1. Introduction

Currently, 55% of the world's population lives in cities. This is expected to reach 68% by 2050 and then continue to increase [1]. The cause of this extraordinary growth is twofold: population growth and displacement of the rural population to urban areas, even from intermediate cities to larger ones. To accommodate this increase in population, cities must grow physically and functionally. They can do this in two ways: expanding or densifying.

The surface expansion of a city implies the occupation of natural or agricultural land through low-density and energy-inefficient developments, which make the provision of environmental infrastructure and social services to the population more difficult. Thus, the sustainability of this form of growth is doubtful [2–4].

In contrast, a compact city occupies less open land and allows multiple uses of the space, resulting in higher energy efficiency and lower water consumption. Consequently, densification can be considered as more environmentally sustainable than expansion. However, this claim is currently being questioned because densification implies difficulties in providing the city with a properly connected and effectively distributed green infrastructure, and this problem worsens when important consequences such as the effects of climate change are considered [5].

All forms of urban growth exacerbate the already serious environmental problems of cities [6]. There are three fundamental causes of its unsustainability: (1) the occupation of natural or agricultural land; (2) the large amount of resources (water, energy, etc.) extracted from an extra-urban environment; and (3) the effluents they emit, all of which have a major

impact on climate change [7,8]. At the same time, life in cities is negatively affected by the effects of climate change, and the situation is expected to get worse with current urban growth [9].

In summary, urban areas are critical points that contribute significantly to global warming and are also affected by climate change [10–12]. Thus, the magnitude of the inevitable growth of cities and the way in which it occurs are global concerns.

One of the measures to move toward urban sustainability and to reduce the effects of climate change is the development of urban green infrastructures (UGIs) [13,14]. Green infrastructure is a term that has multiple definitions. For this case, we adopt the EU's definition of green infrastructure as "*Strategically planned network of high quality natural and semi-natural areas with other environmental features, which is designed and managed to deliver a wide range of ecosystem services and protect biodiversity in both rural and urban settings*" [15]. In urban areas, many different features may be part of green infrastructures (e.g., parks, gardens, allotments, community gardens, cemeteries, green roofs, urban orchards, etc.) as far as they are part of an interconnected network and are delivering multiple ecosystem services [16,17].

Urban green infrastructures are being increasingly recognized as key providers of ecosystem services (ES) in cities, which is crucial, given its relevant role in promoting resilience and quality of life in cities, as well as urban sustainability [18–20]. For that reason, the importance of urban ES is readily acknowledged by scientists, but there is still a lack of support from urban planners [21].

Increasing the recognition of the importance of ES provided by UGIs, and obtaining the support of urban planners, require quantitative assessments, as they lead to credible and more realistic assessments of the ES provided by UGIs [22,23].

The final objective of this paper is to highlight the potential UGIs for sustainable urban planning based on their capacity to mitigate climate change [24,25]. Therefore, in this paper we study the $CO_2$ mitigation potential of UGIs through a multi-intervention local case in the peri-urban surroundings of a big European city. We consider two inseparable aspects: the amount of atmospheric $CO_{2\text{-eq}}$ reduced through direct carbon uptake of the UGI and also the emission of greenhouse gases (GHGs) due to its implementation and maintenance [26]. The different agricultural and forestry interventions analyzed allow us not only to calculate its climate change mitigation potential, but also to distinguish the ecological balance of each one, as an indicator to determine the sustainability of the UGI actions implemented.

## 2. Materials and Methods

### 2.1. Case Study

The study area is located in the peri-urban area of the city of Madrid, covering 200 hectares distributed on both banks of the Manzanares River (Figure 1).

Madrid is the capital of Spain, a country in the south of Europe. Madrid has 6.642 million people, occupies 604 square kilometers, and is divided into 21 districts. The Manzanares River crosses the city of Madrid from the northwest to the south, along a 30 km stretch. The river is home to different ecosystems, running through areas of great environmental value. The course of the Manzanares River is the result of decades of channelling and damming work, although, in 2016, the level of the river was returned to the original flow by opening regulating floodgates, which increased biological diversity. One of the responsibilities of the Madrid City Council is to look after its waters and banks as they flow through the city.

The peri-urban environment in which the area is located is the origin of the environmental degradation that affects it. This degradation can be seen in the presence of a high density of infrastructures and installations within or surrounding the area (electricity lines, high density roads, railroad tracks, electric substations, water treatment plants, rubbish dumps, etc.). There is also waste everywhere in the area: rubble, piled up earth, scrap metal, etc.

The study area has been confined between infrastructures and significant activities, at the speed of very fast urban processes, and is progressively degrading, as is usual in the free spaces of peri-urban locations.

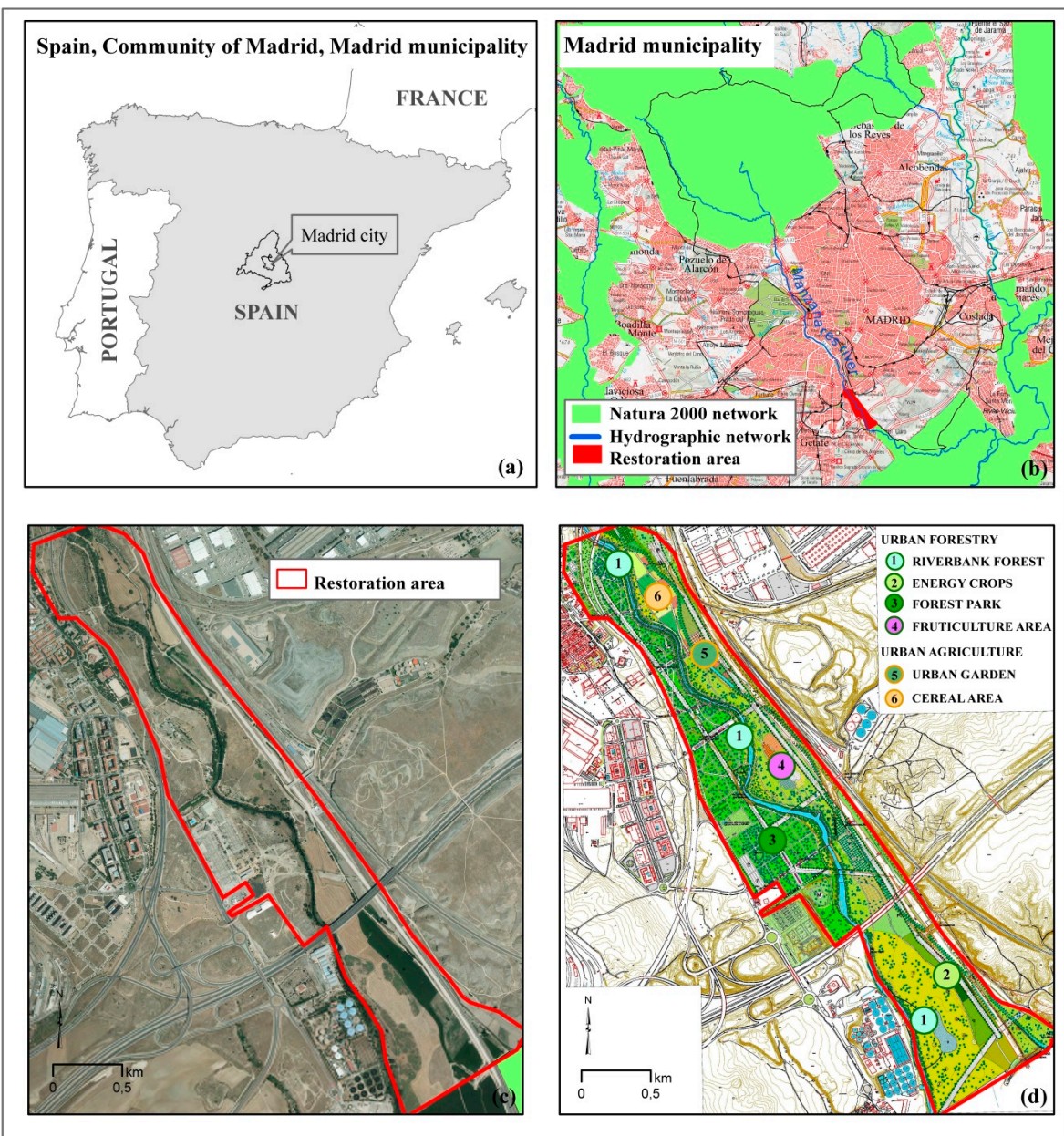

**Figure 1.** Visual information about the project: (**a**) Location of the project in the Community of Madrid and Spain; (**b**) Framework in which the project is inscribed in the municipality of Madrid, where the Manzanares River functions as a connector for protected natural areas (Natura 2000 network); (**c**) pre-operational situation; (**d**) postoperacional situation. Source: (**a–c**) Municipal and regional limits, National Cartographic Base at scale 1:200,000, Orthophoto mosaics of the National Plan for Aerial Orthophotography. National Geographic Institute, Government of Spain (**d**) Own elaboration from the construction project.

The study area has been confined between infrastructures and significant activities, at the speed of very fast urban processes, and is progressively degrading, as is usual in the free spaces of peri-urban locations.

The Madrid City Council proposed the recovery using an approach adapted to the characteristics of an area that still conserves some agricultural activity and valuable natural

and cultural elements that can be enhanced. Agricultural activity is in decline in the municipality of Madrid, and the stronghold that the area presents has traditional and cultural value. An attempt has been made to highlight its ecological, landscape, historical, and cultural values, as well as to respond to social expectations.

The aim was to reach the self-sufficient and environmental sustainability of the area. The project focused on maximizing the efficiency in terms of resource consumption, minimizing the outflows (waste, discharges, and emissions), and optimizing the recirculation flows. An additional requirement from the City Council was to reduce the maintenance cost of the UGI as much as possible.

Thus, the goal was to create a new UGI while restoring the Manzanares riverbank forest and the surrounding area. This was an opportunity for rediscovering this forgotten area of the city and highlighting the important role of the Manzanares river corridor and agricultural activity that still remains today.

The Manzanares restoration project includes different actions with the objective of increasing urban resilience and sustainability: (i) restoration of the riverbank forest; (ii) demonstration of urban biomass production; (iii) creation of a forest park mainly with native tree, bush, and shrub species representative of Madrid habitats; (iv) creation of a didactic area for fruticulture; (v) establishment of urban edible gardens with native horticultural varieties from Madrid, and (vi) establishment of a cereal area for landscaping, cultural, and educational purposes. Description of these actions are shown in Table 1.

### 2.2. Methodology

A city's ecological footprint [27] is the biological productive area required to produce the resources used, and to assimilate the waste generated, by a defined population at a specified standard of living [28]. In other words, a city's ecological footprint is the non-urban space required by a city to continue to exist. This concept shows how cities cannot live without the countryside, something that does not happen the other way around; however, those are complementary spaces where, ideally, synergies would be searched for in order to improve everybody's quality of life.

To calculate it, the resources consumed by a certain population or a certain activity are counted, and the number of hectares of ecologically productive surface necessary to compensate this impact is calculated [29]. This productive surface area is related to the existing areas in each region, which could be oceans, forests, fields, cultivated areas, etc., and their productive potential.

This indicator of sustainability [30] is necessary in the definition of the ecological balance, which is determined by the difference between the ecological footprint and the biocapacity of the territory [31]. When an activity footprint is greater than biocapacity, it is reported to be engaging in ecological overshoot.

The ecological footprint and biocapacity accounts are usually measured in global hectares (gha). These hectares allow researchers to report both the biocapacity of the earth or a region and the demand on biocapacity (the ecological footprint) [32].

In order to calculate the ecological balance of the different actions planned for the implementation and maintenance of the UGI, the authors followed a six steps methodology:

1. Identification of the work units required for the implementation and maintenance of the different types of intervention, agricultural, and forestry that correspond to the UGI. The work units have been obtained from the construction project, which also included the maintenance work.
2. Identification of emission factors of the sources involved in the work units (Table 2).
3. Calculation of the carbon equivalent emissions of the work units using Equation (1).

$$t\ CO_{2\text{-eq}} = \text{Quantity of product} \times \text{Emission Factor} \tag{1}$$

4. Calculation of the ecological footprint. For this, the mass of $CO_{2\text{-eq}}$ has to be converted into global hectares. An equivalence factor is needed that converts a specific land

type (such as cropland or forest) into the universal unit of biologically productive area, a global hectare (gha) [32].

**Table 1.** Description of the urban green infrastructure actions under study.

| Type of Intervention | Action | Description | Area (ha) |
|---|---|---|---|
| Urban forestry | Riverbank forest | Dense and stratified riverbank forest dominated by native species: 2282 *Populus nigra,* 1938 *Fraxinus angustifolia*, 1322 *Populus alba*, 1144 *Ulmus pumila*, 749 *Tamarix gallica*, 489 *Salix alba*, 483 *Cercis siliquastrum*, 458 *Acer pseudoplatanus*, 300 *Salix eleagnus*, 300 *Salix fragilis*, 300 *Salix purpurea*, 260 *Platanus hibrida*, 254 *Salix tortuosa*, 250 *Ulmus minor*, 214 *Gleditsia triacanthos*, and 24 *Juniperus commmunis*. The presence of a strip of riparian forest next to the riverbed will be extremely beneficial from an ecological and landscape point of view. | 72.4 |
| | Energy crops | The species dedicated to biomass production have been selected on the basis of the following criteria: having high levels of biomass productivity with low production costs, possibility of developing on marginal land and low requirement for conventional agricultural machinery: 10,000 *Populus x euroamericana*, 10,000 *Salix* spp., 10,000 *Robinia pseudoacacia*, and 10,000 *Ulmus minor* planted. Twelve years after planting, all trees will be cut for energy use. | 9.5 |
| | Forest Park | Forest park with native tree, bush, and shrub species representative of Madrid habitats: 4375 *Pinus pinea*, 1908 *Junglans regia*, 1000 *Populus nigra*, 933 *Prunus dulcis*, 905 *Arbutus unedo*, 846 *Celtis australis*, 767 *Ulmus minor*, 740 *Quercus faginea*, 740 *Quercus ilex*, 700 *Platanus hispánica*, 399 *Acer pseudoplatanus*, 368 *Mimosa floribunda*, 350 *Cupressus sempervirens*, 330 *Fraxinus angustifolia*, 275 *Betula pendula*, 260 *Philadelphus coronaries*, 260 *Ginkgo biloba*, 220 *Robinia pseudoacacia*, 120 *Cedrus atlántica*, and 88 *Magnolia grandiflora*. | 33.2 |
| | Fruticulture area | Didactic area through the cultivation of ornamental fruit trees and shrubs: 9734 *Pyrus communis*, 7470 *Olea europea*, 4000 *Malus domestica*, 3470 *Prunus domestica*, 1390 *Corylus avellana*, 1210 *Prunus amygdalus*, 100 *Arbutus unedo*, and 496 *Prunus avium plena*. | 60.0 |
| Urban agriculture | Urban edible gardens | Organic agriculture and native varieties of the Community of Madrid. Most relevant species to plant: tomatoes (*Solanum lycopersicum*), asparagus (*Asparagus officinalis*, L.) and strawberries (*Fragaria vesca*) of Aranjuez, Sierra beens (*Phaseolus coccineus* L.), Chinchón garlic (*Allium sativum*), etc. | 3.7 |
| | Cereal area | Typical crop rotation in dry lands for landscape, cultural, and educational purposes: cereal—sunflower (*Helianthus* L.)—cereal—grain legumes—cereal—fallow land. Cereals will be wheat (*Triticum aestivum*), barley (*Hordeum vulgare*), and rye (*Sacale cereal*); legumes, chickpeas (*Cicer arietinum*), lentils (*Lens culinaris*), chickling vetch (*Lathyrus sativus*), and vetches (*Vicia sativum*). | 6.5 |

**Table 2.** Emission factors [33,34].

| Emission Source | Emission Factor |
|---|---|
| Diesel | 2.79 kg $CO_{2\text{-eq}}$/L |
| Fertilizer | 2 kg $CO_{2\text{-eq}}$/Kg |
| Pesticides | 8 kg $CO_{2\text{-eq}}$/Kg |
| Potable irrigation water | 395 g $CO_{2\text{-eq}}$/m$^3$ |
| Vegetal residues | 0.15 t $CO_{2\text{-eq}}$/t$^1$ residues |

The Intergovernmental Panel on Climate Change Guidelines for National Greenhouse Gas Inventories [35] establishes, in 200 tons of dry matter per hectare, the amount of carbon

present in a temperate ocean forest in Europe. Additionally, IPCC give factors of 0.5 and 3.6 to convert the amount of dry biomass into the amount of carbon, and convert this, in turn, to $CO_{2\text{-eq}}$. Therefore, the amount of t $CO_{2\text{-eq}}$ in 1 ha of temperate oceanic forest is known: 360 t $CO_{2\text{-eq}}$/ha. This factor allows us to estimate the equivalent area of forest needed to compensate the emissions derived from the implementation and management of UGI plantations. The equivalence factor for forest, 1.34 [36], is used to calculate the global hectares (gha).

5.  Calculation of $CO_{2\text{-eq}}$ absorption and biocapacity of the different interventions. The annual $CO_2$ capture rate by urban forestry action has been calculated based on a methodology of the Spanish Ministry of Agriculture, Food, and Environment [37]. This methodology calculates the $CO_{2\text{-eq}}$ absorptions of new forest plantations in line with the Intergovernmental Panel on Climate Change guidelines and good practice guidance.

The methodology allows an ex-ante calculation, i.e., before planting, based on the estimation of the species growth for the duration of the project. In this way, it is possible for the project promoters to know what the absorptions that their project will achieve will be. The calculation is based on the determination of carbon dioxide absorptions per planted specimen, and this data is then applied to the whole project, depending on the number of specimens expected at the end of the period of permanence (20, 25, 30, 35, and 40 years).

This period depends on the management carried out. When the management is not commercial, the horizon is 40 years, but when the plantation has a commercial purpose, the $CO_{2\text{-eq}}$ absorbed is estimated at the end of the rotation period, which depends on the specific species.

According to this guide, the estimated absorptions of the species selected in the project vary between 0.05 and 1.3 t $CO_2$/specimen over a 40-year horizon.

The annual $CO_2$ capture rate by the crops of the urban agriculture action has been calculated based on the biomass production data and its carbon content [38]. Based on Mota et al. [38], and taking into account the crops included in the urban agriculture action, the urban garden action capture 16.32 t$CO_{2\text{-eq}}$/ha per year and the cereal area capture 13.45 t$CO_{2\text{-eq}}$/ha per year, considering a time horizon of 40 years.

In order to compare the project's ecological footprint and its ecological balance, the total $CO_{2\text{-eq}}$ absorption values have been converted into bioproductive capacity in global hectares (gha).

6.  Calculation and analysis of the ecological balance. Ecological balance is the difference between the ecological footprint and the biocapacity of the action. If the balance is positive, the biocapacity of the UGI compensates the footprint demanded by the UGI itself in its establishment. This implies that the establishment of the UGI has a positive ecological impact and will contribute to the mitigation of climate change.

## 3. Results

Like any other activity, the implementation and maintenance of the forestry and agricultural actions included in the development of a UGI are a source of GHG emissions (Tables S1–S6). These emissions depend on the way in which both the implementation of the vegetation and its maintenance is done.

Figure 2a show that emissions released during implementation and management activities of the agriculture actions are much higher that the forestry actions in a time horizon of 40 years. Agriculture requires annual works and inputs, which are not necessary in the case of forestry plantations, especially those that try to replicate the natural ecosystem (riverbank forest).

Additionally, agriculture demand inputs every season, and in the case of urban edible gardens, water is also needed for irrigation. Meanwhile, in the case of tree plantations, because the project is in a valley area and native species have been selected, it has only been necessary to consider root irrigation in the plantation and a second support irrigation in the course of the first year.

Figure 2b graphically shows how a large amount of these emissions comes mainly from machinery, but also it also comes from inputs: mostly fertilization and planned phytosanitary treatments.

From the total emissions, 59% come from the use of machinery. Actions during planting, such as soil tilling, mechanical opening of holes, or irrigation, are carried out with machinery whose consumption of fuel is high, which leads to a high emission of GHG.

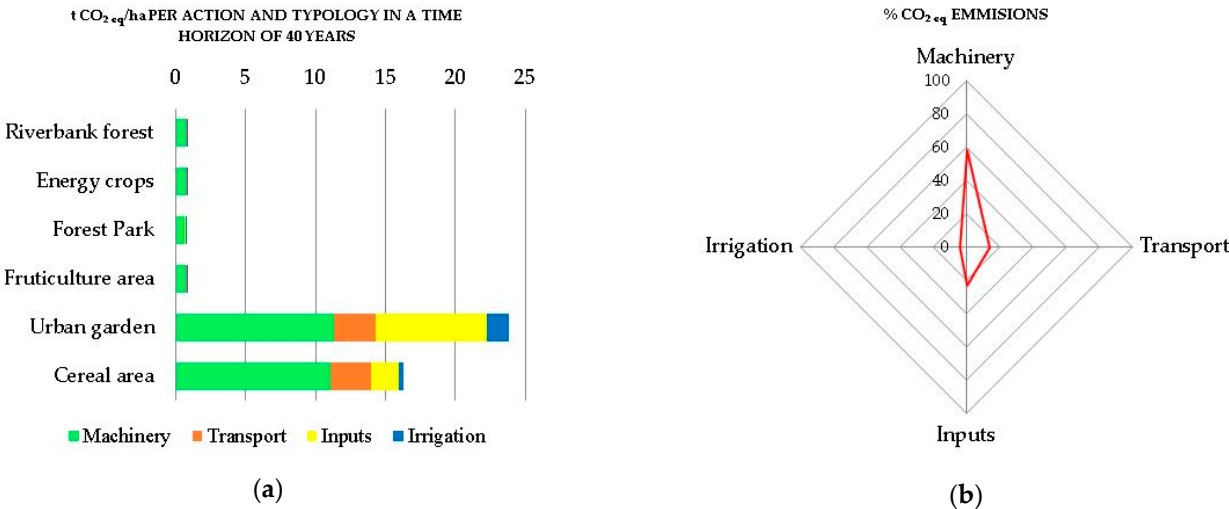

(a)　　　　　　　　　　　　　　　　　　　　　　　　(b)

**Figure 2.** $CO_{2\text{-eq}}$ emission due to implementation and maintenance of the forestry and agricultural actions included in the development of the urban green infrastructures (UGI) in a time horizon of 40 years: (**a**) t $CO_{2\text{-eq}}$/ha per action and typology; (**b**) % $CO_{2\text{-eq}}$ emissions per action.

In the case of inputs, a high percentage of 23% is due to the use of fertilization in urban agriculture areas.

Much less significant in terms of emissions, transport account for 14% of total emissions. This low percentage is due to the minimizing of the number of trips by selecting suppliers close to the plantation site and also through consistent management of the works.

The percentage of emissions due to irrigation is also low, 4%. This is due to the selection of autochthonous species with low water requirements and the location of the project on the river.

The analysis in Figure 3a shows that the ecological balance of the UGI is positive and, therefore, the evaluated UGI contributes to climate change mitigation. The total ecological balance is equal to 73.77 gha, derived from a total ecological footprint of the implementation and management of the UGI of 1.22, compared to the total biocapacity of 74.99 gha. This is because all the types of actions considered, except for energy crops, have a positive balance (Figure 3b).

In the case of energy crops, the balance is slightly negative. These energy crops are harvested after 12 years of planting. The $CO_2$ absorbed during the time the crop is planted will be emitted when it is burned to produce energy. This is the reason for such a negative balance.

Figure 3 illustrates two additional results. The first one is the different behavior between agricultural and forestry actions. Biocapacity of herbaceous species is lower than that of forest species. This is because agricultural species are herbaceous, and therefore have a lower aerial and subterranean biomass, and are harvested periodically. Consequently, the effectiveness of urban agriculture action in the storage of carbon per hectare is lower than that of any of the other forestry actions.

Nevertheless, as long as growth is high, agricultural crops are carbon sinks and, therefore, urban agriculture is, much like tree plantations, an effective mechanism to mitigate the increase of atmospheric $CO_2$ [38].

Secondly, the figure also shows that reforestation of natural areas is more effective in storing carbon than parks, tree cultivation, or other urban green areas.

**Figure 3.** Ecological balance: (**a**) Per UGI; (**b**) Per type of action that conform the UGI.

## 4. Discussion

The actions included in the development of the UGI imply the uptake of 106 t C ha$^{-1}$ (2.67 t C ha$^{-1}$ yr$^{-1}$). Other studies concerning climate mitigation of urban vegetation estimate this value as being in a similar range.

In the US, the overall carbon storage of urban tree cover among 28 cities was 76.9 t C ha$^{-1}$, with the net carbon sequestration rate 2.05 t C ha$^{-1}$ yr$^{-1}$; but it varied between 31.4 t C ha$^{-1}$ for South Dakota and 141.4 t C ha$^{-1}$ for Omaha [39].

In China, C storage by Hangzhou's urban forests was estimated at 30.25 t C per hectare and 1.66 t C ha$^{-1}$ yr$^{-1}$ as the average carbon storage and sequestration rate [40].

In Spain, the net carbon sequestration rate in the municipality of Barcelona was estimated at 1.24 t ha$^{-1}$ of urban green that included urban parks, lawns, allotment gardens, permanent crops, and flowerbeds [41].

As we have established before, the differences can be explained by the type of vegetation, composition, or age [42], or by the type of management [43,44].

## 5. Conclusions

The implementation of any type of UGI in the city is positive from a climate change mitigation standpoint, but its effectiveness will depend on the chosen typology. Taking advantage of degraded areas that often characterize the surroundings of the city to recover natural spaces, such as riverbanks or native forests, or to recover and enhance agricultural activity, is a good option for counteracting urban warming and contributing to urban sustainability. This is particularly true if we compare it with other possible uses of those spaces, such as residential, services, commerce, offices, and industry, that have more obvious and immediate profitability but which contribute to urban warming and climate change.

The analysis carried out has shown the benefits of a series of interventions that configure an UGI to contribute to the mitigation of climate change. The results show that the absorption capacity of the new UGI is much greater than its ecological footprint and continues to mitigate emissions from other urban activities, thus improving urban sustainability.

An increasing number of cities are supporting this type of action, driven by social interest and given the magnitude of foreseeable urban growth and the way in which it is done. Growth by urban expansion consumes a large amount of open space, often of environmental, landscape, or productive interest. The key to favoring urban sustainability through green infrastructure in this case would be to locate the areas of urban expansion over areas of less environmental value, integrating the areas of greater environmental value within the green infrastructure.

Growth by densification consumes less open space but shows weaknesses in reserving sufficient land, adequately distributed, to install significant green infrastructure from the point of view of urban sustainability and the reduction of its carbon footprint. On the other hand, there is an urban pressure on certain green spaces, for example, those existing between the buildings that constitute open blocks and the interior courtyards in closed blocks.

These effects translate into a deficit of green infrastructure. The key to promoting sustainability in this case lies in urban planning and would consist of giving green infrastructure the same importance as the rest of the basic infrastructure: water, sanitation, energy, and transport, providing urban plans with sufficient space to implement it. In addition, compact cities have another opportunity: the possibility of taking advantage of peri-urban spaces to locate green infrastructure and, in particular, those degraded by the expansion or operation of the city itself. Finally, we should mention the growing sensitivity to urban agriculture and its relevant role in urban infrastructure, which calls for planning to consider it as another urban activity.

It is therefore possible to incorporate green infrastructure into the foreseeable growth of cities, whether this is oriented towards low-density extensification over the environment or towards densification in compact cities. Nevertheless, UGIs still suffer from insufficient consideration for urban planning and management. This fact seems to reveal a weakness in the scientific approaches aimed at facilitating the creation of significant green structures from the point of view of urban sustainability and resilience. The analysis carried out demonstrates, in a quantitative way, its contribution to the mitigation of climate change.

It is necessary to move towards the implementation of green infrastructures in cities that are planned, systematic, effective, efficient, and have significant effects on urban sustainability. This requires that society, urban authorities, and more specifically, urban planners consider the green grid as an urban infrastructure whose structure and urban functionality is important, and also view it in terms of image, prestige, and urban sustainability. To get this support, it is essential that scientific research in the field quantifies these benefits, based on real experiences in the city.

**Supplementary Materials:** The following are available online at https://www.mdpi.com/2073-4395/11/1/31/s1, Table S1. Riverbank forest: imple-mentation and maintenance activities; Table S2. Energy crops: implementation and maintenance activities; Table S3. Forest park: implementation and maintenance activities; Table S4 Fruticulture area: implementation and maintenance activities; Table S5 Urban edible garden: implementation and maintenance activities; Table S6 Cereal area: implementation and maintenance activities

**Author Contributions:** Conceptualization, L.R.-G. and M.T.G.-V.; methodology, L.R.-G. and M.T.G.-V.; software, M.T.G.-V., M.G.V., and L.R.-G.; validation, M.T.G.-V., M.G.V., and L.R.-G.; formal analysis, M.T.G.-V., M.G.V., and L.R.-G.; investigation, M.T.G.-V. and L.R.-G.; resources, M.T.G.-V., M.G.V., and L.R.-G.; data curation, M.T.G.-V., M.G.V., and L.R.-G.; writing—original draft preparation, M.T.G.-V., M.G.V., and L.R.-G.; writing—review and editing, M.T.G.-V., M.G.V., and L.R.-G.; visualization, M.T.G.-V., M.G.V., and L.R.-G.; supervision, M.T.G.-V., M.G.V., and L.R.-G. All authors have read and agreed to the published version of the manuscript.

**Funding:** This research received no external funding.

**Institutional Review Board Statement:** Not applicable.

**Informed Consent Statement:** Not applicable.

**Data Availability Statement:** Data is contained within the article or supplementary material.

**Acknowledgments:** The authors want to acknowledge the City Hall of Madrid, for its support in the development of this study.

**Conflicts of Interest:** The authors declare no conflict of interest.

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
