# Peer review of "Implementation of Urban Green Infrastructures in Peri-Urban Areas: A Case Study of Climate Change Mitigation in Madrid"

_agronomy, doi:10.3390/agronomy11010031_

Round 1
Reviewer 1 Report
The paper is interesting and touches on very current problem of the idea of development of urban green infrastructure in dense urban structures. I appreciate several things, especially the concept of the conducted research and the idea to calculate ecological balance of the area using parameters of calculated ecological footprint and biocapacity using practical example of revitalized area of peri-urban Madrid.
Nevertheless I suggest the few following revisions.
Lines 26-27 – I suggest using more precise keywords, as they do not match perfectly to the submitted paper. I suggest referring to the analyzed area.
Lines 39-44 – Authors discuss the problem of compact or low-density city, but it the end they do not refer to the problem. I suggest adding a point of Discussion where it would be possible to conclude and to summarize the results in order to clearly define their point of view in this topic.
Line 81 – some more detailed information about the form of collecting data would be required.
Lines 93-94 – There is a lack of reference to the source materials (maps, aerial photos etc.) of the illustration with adding information about copyrights. The illustration, although very important, is blurred and should be more carefully prepared, with adding subtitles or extra numbers (a) to (d).
Line 191 – I suggest dividing the point into two separate parts. The discussion should place the discussed case study in a wider context of similar implementations in the world (or maybe also in Madrid itself) to add some comparative analysis with different but similar solutions and how they are made. This part should refer to the wider context and concepts of implementation.
Lines 282-432 – for quite short text the number of references is disproportionately large. I suggest, especially in the Introduction, to choose only the ones strictly referring to the topic of the paper and not adding everything connected with the subject. If referring to larger numbers of references at one point I would suggest adding some more precise information and break it into more detailed sentences, as each and every position in the bibliography do not refer strictly to the same topic, ex. lines: 34,38, 48, 50, 56, 63, 65.
Reviewer 2 Report
It is an outstanding paper that deals with a current subject and incorporates an interesting vision to include urban green infrastructures in peri-urban areas. The article responds adequately to a scientific structure and provides relevant results. Nevertheless, it could be improved including a significant number of current references (last five years).
